# Cytokine-induced megakaryocytic differentiation is regulated by genome-wide loss of a uSTAT transcriptional program

Hyun Jung Park[1,2], Juan Li[1,2], Rebecca Hannah[1,2], Simon Biddie[1,2], Ana I Leal-Cervantes[1,2], Kristina Kirschner[1,2], David Flores Santa Cruz[1,2], Veronika Sexl[3], Berthold Göttgens[1,2,*] & Anthony R Green[1,2,4,**]

## Abstract

Metazoan development is regulated by transcriptional networks, which must respond to extracellular cues including cytokines. The JAK/STAT pathway is a highly conserved regulatory module, activated by many cytokines, in which tyrosine-phosphorylated STATs (pSTATs) function as transcription factors. However, the mechanisms by which STAT activation modulates lineage-affiliated transcriptional programs are unclear. We demonstrate that in the absence of thrombopoietin (TPO), tyrosine-unphosphorylated STAT5 (uSTAT5) is present in the nucleus where it colocalizes with CTCF and represses a megakaryocytic transcriptional program. TPO-mediated phosphorylation of STAT5 triggers its genome-wide relocation to STAT consensus sites with two distinct transcriptional consequences, loss of a uSTAT5 program that restrains megakaryocytic differentiation and activation of a canonical pSTAT5-driven program which includes regulators of apoptosis and proliferation. Transcriptional repression by uSTAT5 reflects restricted access of the megakaryocytic transcription factor ERG to target genes. These results identify a previously unrecognized mechanism of cytokine-mediated differentiation.

**Keywords** cytokine; differentiation; haematopoiesis; JAK/STAT; megakaryocyte

**Subject Categories** Immunology; Signal Transduction; Transcription

The EMBO Journal (2016) 35: 580–594

See also: **T Decker** (March 2016)

## Introduction

Metazoan development is controlled by complex gene regulatory networks that need to respond rapidly to extracellular signals (Davidson, 2006, 2010). The JAK/STAT signaling pathway is essential for normal development and adult homeostasis in organisms from *C. elegans* to mammals (Stark & Darnell, 2012; Wang & Levy, 2012). Following their interaction with cellular receptors, many cytokines and other growth factors trigger rapid activation of JAK family kinases with consequent tyrosine phosphorylation and activation of STATs. Activated STATs (pSTATs) accumulate in the nucleus, bind to DNA, and regulate the transcription of target genes (Levy & Darnell, 2002).

Hematopoiesis is regulated by multiple cytokines which activate the JAK/STAT pathway (Metcalf, 2008) and also by combinatorial transcription factor interactions that establish cell-type specific patterns of gene expression (Orkin & Zon, 2008). However, the mechanisms by which lineage-affiliated transcriptional programs are modulated by JAK/STAT activation remain obscure. Non-canonical modes of JAK/STAT signaling have been described and include the demonstration that JAK2 functions in the nucleus as a histone kinase (Dawson *et al*, 2009, 2012; Griffiths *et al*, 2011), that JAK/STAT signaling in *Drosophila* disrupts heterochromatin (Shi *et al*, 2006, 2008), and that overexpressed STAT mutants that cannot be tyrosine-phosphorylated can act as transcription factors with the best-studied examples coming from STAT1 and STAT3 (Chatterjee-Kishore *et al*, 2000; Yang *et al*, 2005, 2007; Cui *et al*, 2007; Cheon *et al*, 2011). However, none of these insights have linked JAK/STAT activation to the transcriptional networks that control hematopoiesis, and we currently lack a genome-wide understanding of the biological role of any tyrosine-unphosphorylated STAT (uSTAT).

STAT5 exists as 2 isoforms (STAT5A and STAT5B with 95% amino acid identity) encoded by two closely linked genes (Hennighausen & Robinson, 2008) that are essential for normal organogenesis (Teglund *et al*, 1998; Cui *et al*, 2004), hematopoiesis, and lymphopoiesis (Hoelbl *et al*, 2006; Yao *et al*, 2006). Overexpression of constitutively active STAT5A gives rise to multilineage leukemia (Kato *et al*, 2005; Moriggl *et al*, 2005), hematopoietic stem and progenitor cell (HSPC) expansion, and enhanced erythropoiesis (Schuringa *et al*, 2004). Within hematopoiesis, the two isoforms are

1   Cambridge Institute for Medical Research, Wellcome Trust/MRC Stem Cell Institute, Cambridge, UK
2   Department of Haematology, University of Cambridge, Cambridge, UK
3   Institute of Pharmacology and Toxicology, Veterinary University Vienna, Vienna, Austria
4   Department of Haematology, Addenbrooke's Hospital, Cambridge, UK
    *Corresponding author. Tel: +44 1223 336895; E-mail: bg200@cam.ac.uk
    **Corresponding author. Tel: +44 1223 336820; E-mail: arg1000@cam.ac.uk

largely redundant (Liu *et al*, 1997; Udy *et al*, 1997), widely expressed, and frequently referred to collectively as STAT5. Different cytokines activate STAT5 in distinct cell types (Schepers *et al*, 2012). In HSPC and within the megakaryocytic lineage, STAT5 is activated by thrombopoietin (TPO) (Kirito & Kaushansky, 2006; Vainchenker & Constantinescu, 2013) which, together with its receptor MPL, is essential for normal megakaryopoiesis (Gurney *et al*, 1994; Alexander *et al*, 1996; de Sauvage *et al*, 1996; Bunting *et al*, 1997) and normal hematopoietic stem cell (HSC) behavior (Alexander *et al*, 1996; Solar *et al*, 1998; Fox *et al*, 2002; Yoshihara *et al*, 2007). However, little is known about the way in which the consequences of cytokine-induced STAT5 activation are integrated with any lineage-affiliated transcriptional program. Genome-wide analyses have focused on the role of activated pSTAT5 (Yang *et al*, 2011; Dawson *et al*, 2012; Zhu *et al*, 2012; Kang *et al*, 2013), and a general problem for understanding the biology of endogenous uSTAT5 (and all uSTATs) has been the lack of cellular systems in which uSTAT function can be studied separately from the corresponding pSTAT.

# Results

## uSTAT5 is present in the nucleus of hematopoietic stem/progenitor cells

To investigate potential functions for uSTAT5 in hematopoietic stem/progenitor cells, we first explored the subcellular localization of STAT5 proteins in primary $Lin^-Sca-1^+cKit^+$ (LSK) cells isolated from mouse bone marrow. Using an antibody detecting tyrosine-phosphorylated STAT5 (both Tyr 694 in STAT5a and Tyr 699 in STAT5b, henceforth termed pSTAT5), pSTAT5 was weakly detectable in the nuclei of freshly isolated cells, became undetectable after 4 h of serum starvation, and was rapidly induced following TPO stimulation (Fig 1A). By contrast, using an antibody that detects total STAT5 protein, nuclear STAT5 was readily detectable under all three conditions (Fig 1A). These results were confirmed using confocal imaging (Fig EV1A) and indicate the presence of nuclear STAT5 proteins that lack tyrosine phosphorylation (uSTAT5).

We then established an experimental system that would allow separate functional analysis of uSTAT5 and pSTAT5. HPC7 hematopoietic stem/progenitor cells are cytokine dependent, multi-potent, and karyotypically normal (Pinto do *et al*, 1998). HPC7 cells self-renew in the presence of SCF, but undergo megakaryocytic differentiation following TPO stimulation (Figs 1B and EV1B–D) (Dumon *et al*, 2012). We found that TPO robustly induces pSTAT5 in HPC7 cells. Interestingly, however, pSTAT5 disappeared within hours, suggesting that sustained pSTAT5 may not be required for megakaryocyte differentiation in this model (Fig EV1E). Under self-renewal conditions, pSTAT5 was not detectable in either cytoplasmic or nuclear fractions, whereas an antibody to total STAT5 readily detected STAT5 protein in both compartments (Fig 1C, lanes 1 and 2). Even after the removal of both SCF and serum, nuclear STAT5 remained evident in the absence of detectable pSTAT5 (Fig 1D, lane 1). The subsequent addition of TPO resulted in rapid accumulation of nuclear pSTAT5 (Fig 1D). We considered the possibility that even in the absence of SCF and serum, cells might, nonetheless, harbor low levels of nuclear pSTAT5, below the detection threshold for

Western blotting. However, sequential immunoprecipitation of total nuclear STAT5 proteins followed by Western blotting also failed to detect any pSTAT5 in unstimulated cells (Fig 1E, lane 4). Together, these results demonstrate that hematopoietic stem/progenitor cells contain nuclear STAT5 in the absence of detectable pSTAT5 and that HPC7 cells provide a powerful system for dissecting the distinct functions of uSTAT5 and pSTAT5.

## TPO stimulation results in genome-wide redistribution of STAT5 from CTCF to STAT-binding sites

To compare the genomic localization of uSTAT5 and pSTAT5, chromatin immunoprecipitation coupled with high-throughput DNA sequencing (ChIP-Seq) was performed using unstimulated ($T = 0$ min) and TPO-treated ($T = 30$ min) HPC7 cells, respectively. Significant binding events in comparison with control IgG samples were identified using the MACS algorithm (Zhang *et al*, 2008). Using an antibody against total STAT5, we observed a striking redistribution of STAT5 peaks following TPO stimulation (Fig 2A and B). Three distinct clusters of STAT5 peaks were identified. Cluster 1 (4,109 elements) contained peaks present only in unstimulated cells and which therefore represent binding of uSTAT5; this cluster contained multiple genes associated with megakaryocytic differentiation (e.g., Mpl, Gp6, Pf4, and Cd41; Figs 2B and EV2A). Cluster 2 (771 elements) contained peaks present only in TPO-treated cells and which were therefore likely to represent binding of pSTAT5; this cluster included multiple known STAT target genes (e.g., Cish, Bcl6, Pim2, and Socs2, Figs 2B and EV2B). Cluster 3 (198 elements) contained those peaks that were unchanged by TPO treatment. The genomic distribution of STAT5 under those two conditions was also different with a 3-fold enrichment of STAT5 binding in promoter regions in unstimulated cells compared to TPO-stimulated cells ($T = 0$ versus $T = 30$, Fig EV2C). We validated ChIP-Seq data by ChIP-qPCR using antibodies to total STAT5 or pSTAT5. STAT5 binding was confirmed at all 6 target sites examined (Fig 2C and D) and, as expected, pSTAT5 binding was only detectable following TPO stimulation (Fig 2C and D). These results demonstrate that in unstimulated cells, uSTAT5 binds to regions that are completely different to those bound by pSTAT5 following TPO stimulation.

To investigate whether the STAT5 redistribution occurs in a sequence-specific manner, DNA sequences associated with STAT5 peaks were analyzed using MEME (Bailey *et al*, 2009). As expected, the most enriched motif in TPO-stimulated cells (containing pSTAT5) was a STAT consensus-binding site or GAS motif (Fig 2E and Appendix Table S1). In marked contrast, the most enriched motif in unstimulated cells (containing only uSTAT5) was a CTCF-binding site (Fig 2E and Appendix Table S1), which was centrally located within regions bound by STAT5 (Fig EV2D). Our STAT5 ChIP-Seq results were then compared to previously reported CTCF ChIP-Seq data in HPC7 cells (Calero-Nieto *et al*, 2014). Approximately two-thirds (64%) of the uSTAT5 peaks present in unstimulated cells colocalized with CTCF peaks, and the vast majority (97%) of these STAT5 peaks were lost following TPO treatment (Figs 2F and G, and EV2E). To exclude the possibility that the STAT5 antibody recognizes non-specific proteins, we tested antibody specificity using Western blotting and immunofluorescence assays on STAT5-null MEFs. The absence of immunoreactivity in STAT5-null samples clearly demonstrates that the STAT5 antibody used in this study is

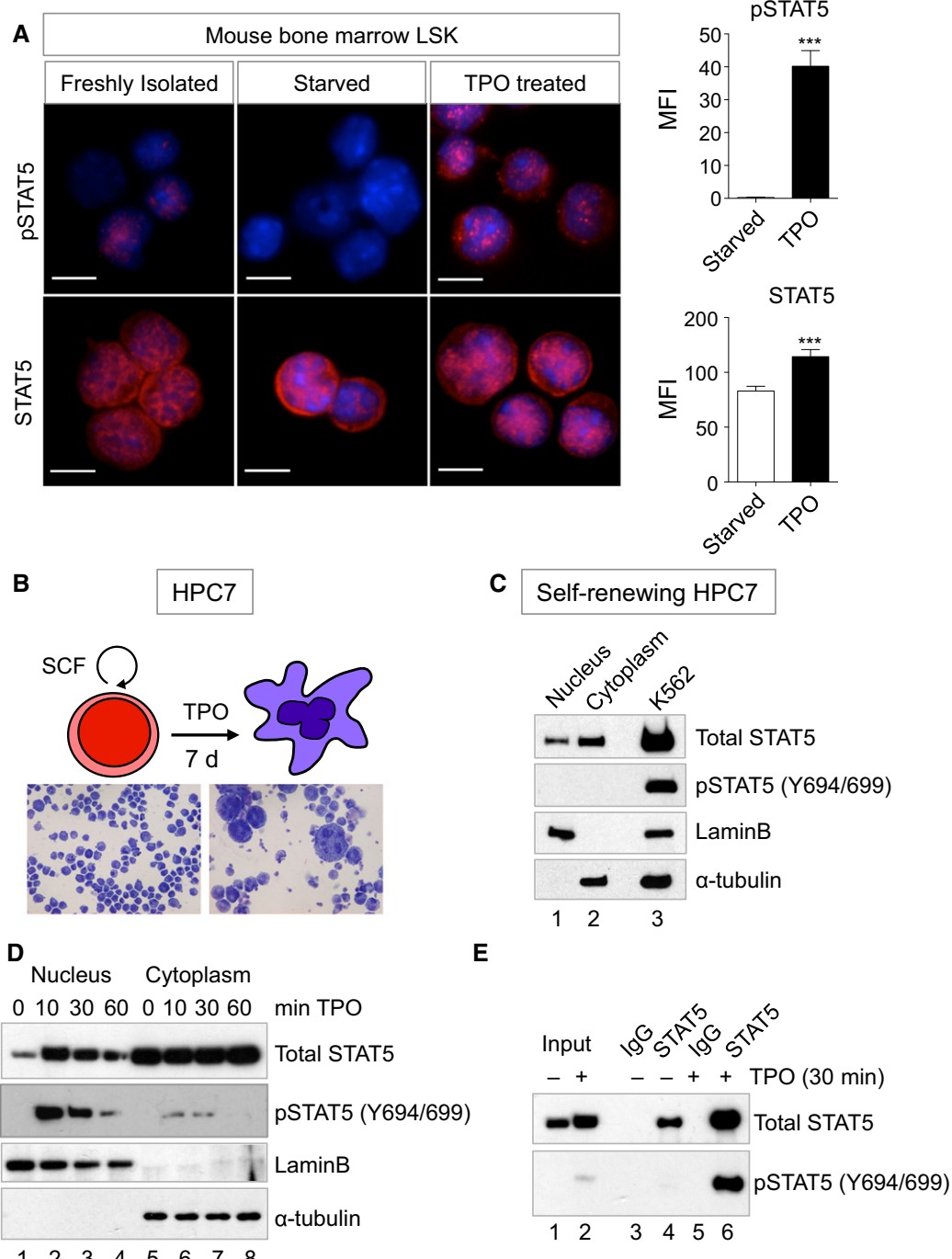

**Figure 1.   STAT5 proteins localize to the nucleus in the absence of tyrosine phosphorylation.**

A   Immunofluorescence staining of Lin⁻Sca-1⁺cKit⁺ (LSK) cells using an antibody specific for phosphorylated STAT5 (pSTAT5) or total STAT5. Scale bars represent 5 μm. Bar graphs showing the quantification of nuclear localization presented as mean fluorescence intensity (MFI). Data are shown as means ± SD of three independent experiments. Two-tailed Student's *t*-test, ***$P$ < 0.001.

B   Diagram showing HPC7 cells that self-renew in the presence of stem cell factor (SCF) and undergo megakaryocytic differentiation in response to thrombopoietin (TPO).

C   Western blot analysis of HPC7 cells grown in the presence of SCF demonstrates nuclear and cytoplasmic uSTAT5 in the absence of pSTAT5. K562 cell lysates were used as a positive control for pSTAT5.

D   Western blot analysis of HPC7 cells after TPO stimulation (100 ng/ml) at the indicated time points.

E   Western blot analysis was performed after immunoprecipitation of STAT5 from unstimulated and TPO-stimulated cells using antibodies to total STAT5 or to pSTAT5 (pY694/p699). 1% (5 μg) of lysate was loaded as input control.

Source data are available online for this figure.

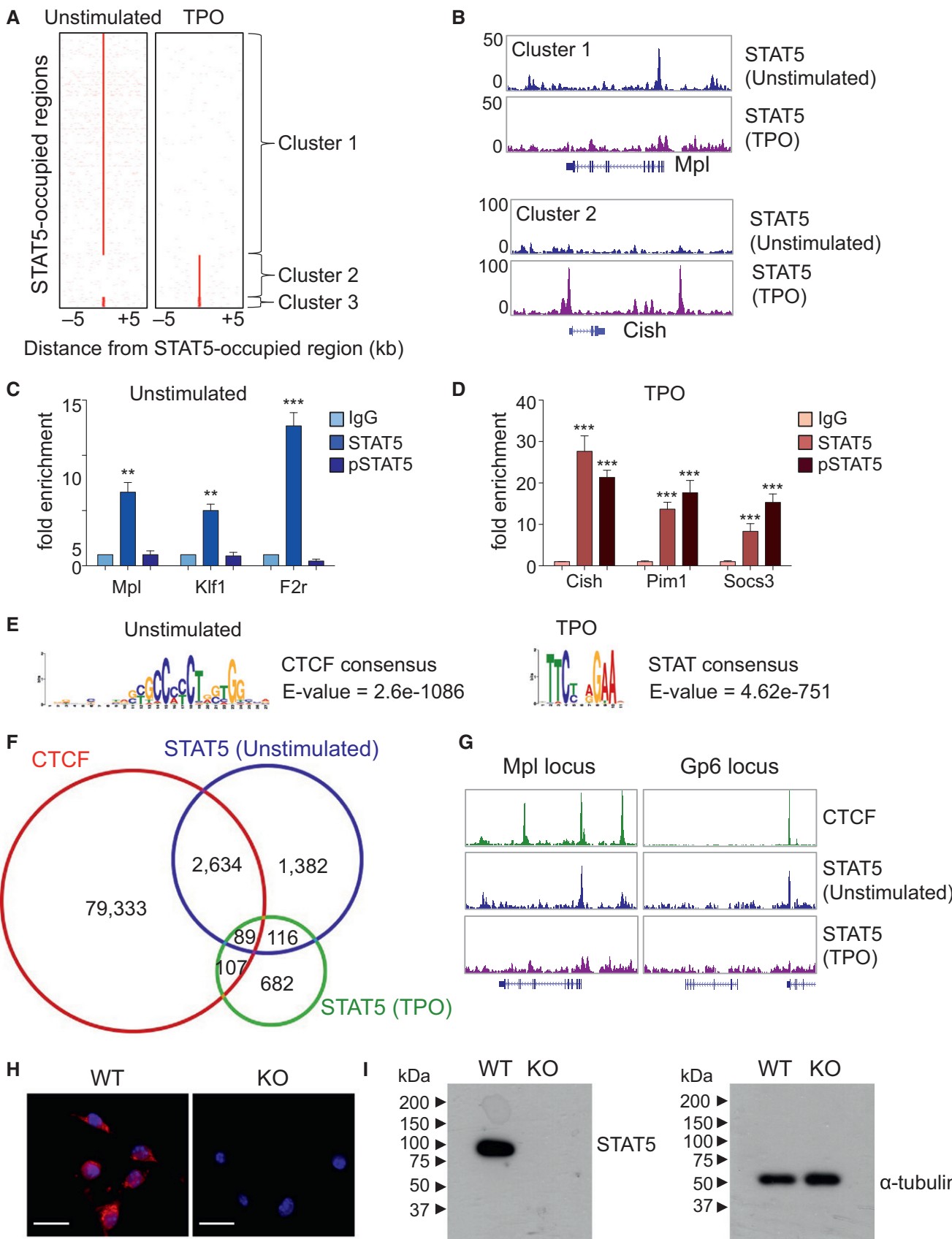

**Figure 2.**

**Figure 2.  TPO stimulation results in genome-wide redistribution of STAT5.**

A      Summary of ChIP-Seq results obtained using HPC7 cells and an antibody to total STAT5. For each STAT5-occupied region, the occupancy of STAT5 in unstimulated ($T = 0$ min) and in TPO-stimulated ($T = 30$ min) cells is indicated within a 5-kb window centered on the total STAT5-bound region.

B      Examples of loci containing ChIP-Seq peaks from cluster 1 (i.e., uSTAT5 peaks) or cluster 2 (i.e., pSTAT5 peaks) showing the loss of the former and appearance of the latter following TPO stimulation.

C, D  ChIP-Seq results were validated by ChIP-qPCR using antibodies against total STAT5 or pSTAT5. Data represented as means ± SD of three independent experiments. Two-tailed Student's t-test; **$P < 0.01$; ***$P < 0.001$.

E      De novo motif discovery using MEME with STAT5-binding sites as input. The most significantly enriched motifs in unstimulated and TPO-treated HPC7 cells are shown.

F      Venn diagram showing the overlap between STAT5 and CTCF binding.

G      ChIP-Seq genome browser views showing colocalization of uSTAT5 binding with CTCF.

H      Immunofluorescence analysis of STAT5 wild-type and knockout MEFs with 4′,6-diamidino-2-phenylindole (DAPI) staining (blue) and staining with anti-STAT5 antibody (red). Scale bars represent 25 μm.

I      Western blot analysis of cell lysates isolated from STAT5 wild-type and knockout MEFs.

specific to STAT5 proteins (Fig 2H and I). Together, these results demonstrate that TPO stimulation is accompanied by a striking redistribution of STAT5 binding with the loss of uSTAT5 binding to CTCF sites and the recruitment of pSTAT5 to consensus STAT sites.

### uSTAT5 represses a transcriptional program associated with megakaryopoiesis

To explore the biological significance of the genomic regions bound by both uSTAT5 and CTCF, we used the GREAT program (McLean *et al*, 2010). GREAT analysis for the 2,723 regions bound by uSTAT5 and CTCF revealed strong enrichment for genes associated with hematopoiesis, especially megakaryocyte and platelet development, whereas the 994 regions bound by pSTAT5 were enriched for a variety of immunological functions (Fig EV3A).

To further investigate the transcriptional consequences of uSTAT5 binding, shRNA-mediated STAT5 knockdown was performed in self-renewing HPC7 cells, which do not express detectable level of pSTAT5 (Figs 1C and 3A). Two different shRNA constructs markedly reduced levels of total STAT5 (Fig 3A) together with protein and transcript levels of STAT5B, but not STAT5A (Fig EV3B and C), indicating that STAT5B is the major STAT5 protein expressed in HPC7 cells. Knockdown of STAT5 resulted in differential expression of 1,029 genes (367 upregulated; 662 downregulated; FDR <0.001; fold change >1.5). Of note, we observed no significant change in the transcript levels of the canonical STAT5 target genes, including Cish and Socs2, indicating that these changes observed are a consequence of depleting uSTAT5.

Several complementary lines of evidence indicate that uSTAT5 depletion resulted in upregulation of a transcriptional program associated with megakaryopoiesis. (i) Of the 40 most highly upregulated genes following STAT5 knockdown, at least 23 genes are reported to be highly expressed in megakaryocytes and platelets (Fig 3B). (ii) Gene set enrichment analysis (GSEA) using microarray data for distinct hematopoietic progenitors (Pronk *et al*, 2007) showed that genes upregulated by uSTAT5 knockdown were significantly enriched in megakaryocytic progenitors (Fig 3C), whereas downregulated genes were more highly expressed in common lymphoid progenitors or pre-granulocyte/monocyte progenitors (Fig EV4A). (iii) Gene ontology (GO) analysis showed that transcripts upregulated by uSTAT5 depletion were enriched in genes involved in blood coagulation and hemostasis (Fig EV4B) and included several megakaryocyte/platelet genes (e.g., Vwf, Gp9, and

Pf4). By contrast, transcripts downregulated by uSTAT5 depletion were enriched in genes involved in lymphocyte activation and differentiation (Fig EV4C).

Approximately 32% (117 of 367) of upregulated transcripts and 31% (203 of 662) of downregulated transcripts were annotated to uSTAT5 peaks, indicating that they represent direct uSTAT5 targets. Altered expression was confirmed by quantitative RT–PCR for 6 of 6 direct target genes tested (Figs 3D and EV4D), including multiple genes implicated in megakaryopoiesis. Moreover, those megakaryocytic genes (e.g., Mpl, Gp9, and Pf4) were also upregulated by TPO stimulation (Fig 3E). The transcriptional response to TPO of uSTAT5 targets was slower and more prolonged than that of canonical pSTAT5 targets (Fig 3E and F).

Together, these results reveal two distinct transcriptional programs, both triggered by TPO and directly regulated by STAT5. One is driven by pSTAT5 and includes canonical STAT target genes that regulate processes including cell survival and proliferation (e.g., Bcl-x and Pim1). The second was previously unrecognized and is regulated by uSTAT5 which represses multiple megakaryocytic genes.

### uSTAT5 inhibits megakaryocytic differentiation

Consistent with our gene expression data, uSTAT5 depletion resulted in increased numbers of cells with megakaryocytic features including increased size, expression of acetylcholinesterase (Fig 4A and B), increased nuclear ploidy (Fig 4C), and increased expression of CD41 and CD61 (Fig 4D). Importantly, these changes were observed despite the continued presence of SCF and absence of TPO, conditions that limit megakaryocytic differentiation. To examine the effects of increasing uSTAT5 levels on megakaryocyte differentiation, we employed a STAT5B mutant (Y699F) that lacks the key tyrosine phosphorylation site (Fig 4E and F). Compared to cells infected with empty vector (EV) or WT, cells expressing the Y699F mutant exhibited attenuated differentiation into megakaryocytes (Fig 4G–I). We considered the possibility that the Y699F mutant might act in a dominant negative manner to inhibit tyrosine phosphorylation or function of endogenous STAT5. However, pSTAT5 levels in cells expressing the Y699F mutant were similar to those in control cells infected with empty vector (Fig 4F). Moreover, transcript levels of pSTAT5 target genes were not altered by expression of the Y699F mutant, suggesting that Y694F did not affect the function of pSTAT5, including dimerization of pSTAT5 and/or pSTAT5

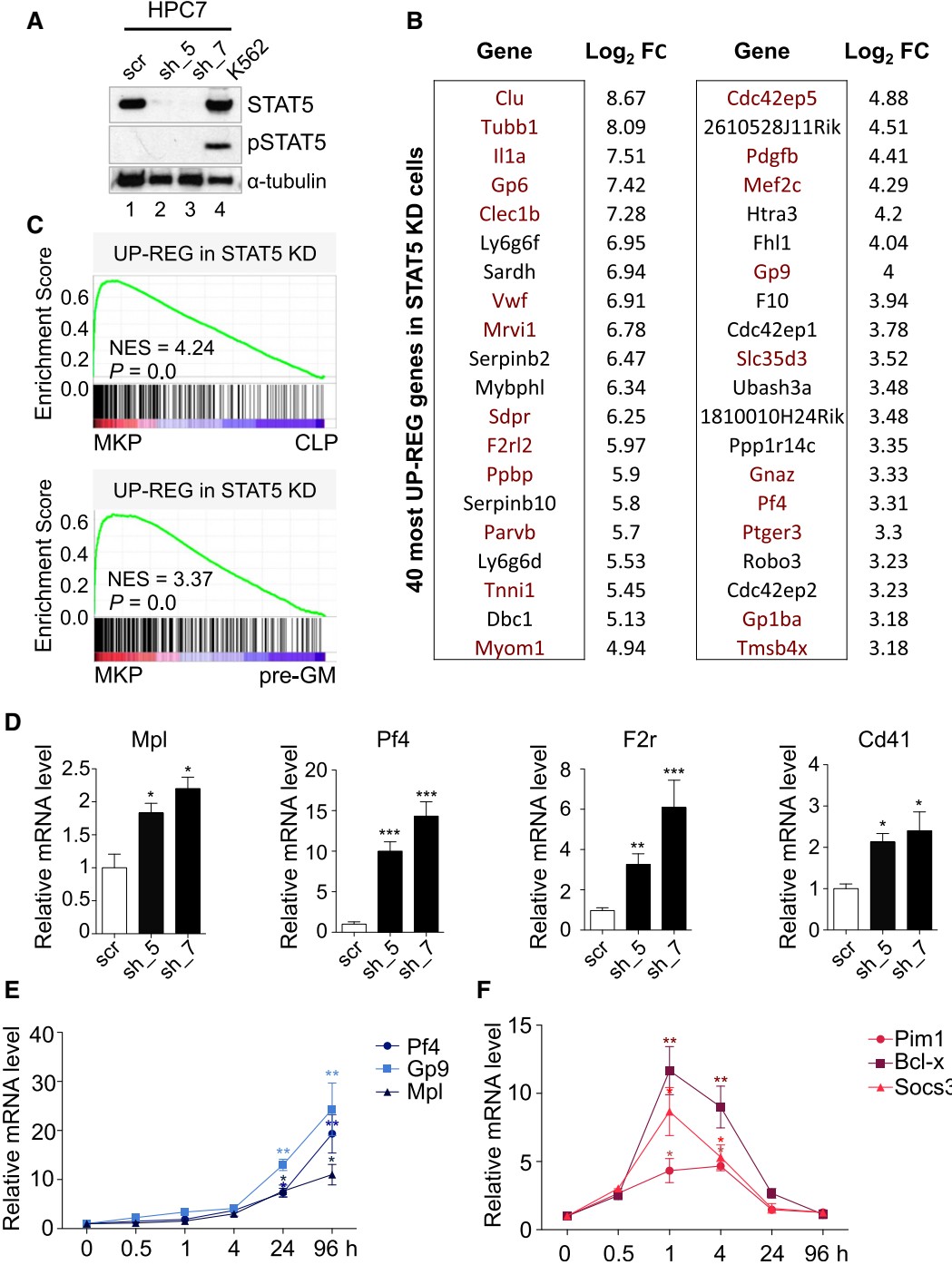

**Figure 3. uSTAT5 depletion activates a megakaryocytic transcriptional program.**

A   Western blot analysis showing an efficient knockdown of uSTAT5 in HPC7 cells on day 3 after infection.

B   RNA-Seq analysis following uSTAT5 depletion by shRNA_7 in HPC7 cells grown in SCF. The 40 most upregulated genes following uSTAT5 knockdown are illustrated. Genes upregulated during megakaryopoiesis are indicated in red.

C   Gene set enrichment analysis showing a significant enrichment of a megakaryocyte progenitor (MKP, Lin⁻Sca1⁻Kit⁺Itga2b⁺Slamf1⁺) signature in STAT5-knockdown cells. CLP, common lymphoid progenitor (Lin⁻Sca1^low^Kit^low^IL7r⁺Flk2⁺); pre-GM, pre-granulocyte/macrophage progenitors (Lin⁻Sca1⁻Kit⁺Itga2b⁻FcgR^low^Slamf1⁻Eng^low^); NES, normalized enrichment score.

D   Quantitative RT–PCR confirmed upregulation of uSTAT5 target genes following uSTAT5 depletion. Relative RNA levels were normalized to 18S rRNA. Bar graphs represent means ± SD of three independent experiments. Two-tailed Student's *t*-test; *$P < 0.05$; **$P < 0.01$; ***$P < 0.001$.

E, F   Kinetics of the upregulation of uSTAT5 (E) and pSTAT5 (F) target genes following TPO stimulation. Mean ± SD; $n = 3$; two-tailed Student's *t*-test; a significant difference from time 0: *$P < 0.05$; **$P < 0.01$.

Source data are available online for this figure.

binding to its target genes (Fig 4J, compare EV with Y699F), whereas multiple uSTAT5 target genes were repressed following introduction of the Y699F mutant (Fig 4K, compare EV with Y699F). Our results therefore indicate that expression of the Y699F mutant inhibited megakaryocytic differentiation by increasing uSTAT5 activity.

To extend these findings to primary cells, lentiviral shRNA constructs were used to deplete STAT5 in mouse bone marrow LSK cells (Fig EV5A). Consistent with our results using HPC7 cells, the depletion of STAT5 was associated with enhanced megakaryocyte differentiation including an increased proportion of cells expressing CD41/CD61 (Fig EV5B) and increased expression of multiple megakaryocyte-affiliated genes (Fig EV5C). Conversely, retroviral expression of the STAT5B Y699F mutant was accompanied by attenuated megakaryocytic differentiation of LSK cells grown in TPO (Fig EV5D and E) and reduced expression of multiple megakaryocyte-affiliated genes (Fig EV5F). Together, these knockdown and overexpression studies demonstrate that uSTAT5 inhibits megakaryocytic differentiation of hematopoietic stem and progenitor cells.

### uSTAT5 antagonizes ERG binding

Combinatorial binding of transcription factors provides a lineage-specific regulatory code. We therefore hypothesized that uSTAT5 might modulate the megakaryocytic transcriptional program by interacting with other transcription factors known to influence megakaryopoiesis. Genome-wide binding of uSTAT5 was compared with ChIP-Seq datasets, generated using HPC7 cells, for 10 key hematopoietic transcription factors (Wilson *et al*, 2010) including 5 implicated in megakaryocytic differentiation (FLI1, ERG, SCL, GATA2, and RUNX1). Clustering of the global occupancy patterns revealed that uSTAT5 binding events were strongly correlated with CTCF and also with ERG, Fli1, and SCL (Fig 5A). Strikingly, approximately 75% of the sites bound by uSTAT5 and CTCF were also occupied by ERG (Fig 5B and Appendix Fig S1A), an ETS family transcription factor known to regulate hematopoietic stem cell self-renewal and megakaryopoiesis (Loughran *et al*, 2008; Ng *et al*, 2011). Moreover, an ETS-binding motif was the second most enriched motif within uSTAT5 peaks

(Appendix Table S1). Using the SPAMO program (Whitington *et al*, 2011), we found that the CTCF and ETS motifs were frequently separated by 7 bp ($P = 8.3e-22$; Appendix Fig S1B), indicating a possible interaction between proteins binding to these two sites.

To investigate whether ERG and uSTAT5 cooperate or compete at shared target sites, ChIP-reChIP experiments were performed to determine whether they bind to the same DNA molecules. Using HPC7 cells grown in self-renewal conditions, chromatin regions precipitated by an antibody to STAT5 were subsequently precipitated by an antibody to CTCF, but not by one to ERG (Fig 5C). Moreover, chromatin regions precipitated by an antibody to ERG were subsequently precipitated by an antibody to CTCF, but not by one to STAT5 (Fig 5D). These results indicate that the observed colocalization reflects mutually exclusive binding of uSTAT5 and ERG to a given CTCF-binding region. Consistent with this interpretation, the depletion of uSTAT5 by shRNA-mediated knockdown in self-renewing HPC7 cells was accompanied by reduced uSTAT5 binding and concomitant increased ERG binding to target sites within the Mpl and F2r loci (Fig 5E and F). There was no alteration of ERG binding at the Med8 locus, an ERG target gene with no colocalization of STAT5 and ERG peaks (Appendix Fig S1C). Importantly, TPO stimulation of HPC7 cells also resulted in reduced binding of uSTAT5 and increased recruitment of ERG at the Mpl and F2r loci (Fig 5G and H). The TPO-induced reduction in uSTAT5 binding remained evident 24 h after TPO stimulation despite the fact that pSTAT5 induction is much more transient (Fig 1D), indicating that the loss of uSTAT5 is not rapidly reversible.

Taken together, these data indicate that prior to the addition of TPO, uSTAT5 antagonizes ERG activity by restricting its access to megakaryocytic target genes and that TPO-induced loss of uSTAT5 binding functions as a molecular switch (Fig 6).

## Discussion

Here, we show for the first time that cytokine-mediated differentiation reflects the loss of a uSTAT transcriptional program. Using a tractable cellular system, which permits analysis of endogenous uSTAT5 function in the absence of pSTAT5, we demonstrate the

---

**Figure 4.  uSTAT5 represses megakaryocytic differentiation.**

A   Schematic diagram of lentiviral uSTAT5 knockdown in HPC7 cells.

B   Depletion of uSTAT5 was accompanied by the appearance of large cells that stain positively for acetylcholinesterase (AchE). Scrambled shRNA (scr), shSTAT5_5 (sh_5), or shSTAT5_7 (sh_7). Scale bars represent 25 μm.

C   Increased percentage of polyploid cells following uSTAT5 depletion.

D   Increased percentage of cells expressing high levels of CD41 and CD61 following uSTAT5 depletion.

E   Strategy for assessing the effect of increasing uSTAT5 on megakaryocytic differentiation of HPC7 cells.

F   Western blot analysis of HPC7 cells expressing empty vector (EV), wild-type STAT5B (WT), or mutant STAT5B (Y699F). EV- and Y699F-expressing cells exhibit similar levels of STAT5 phosphorylation following TPO treatment (100 ng/ml for 30 min).

G   Expression of the Y699F mutant repressed the appearance of large cells that stained for acetylcholinesterase (AchE). Scale bars represent 25 μm.

H   The Y699F mutant increased the percentage of polyploidy cells.

I   The Y699F mutant increased the percentage of cells expressing high levels of CD41 and CD61.

J   Quantitative RT–PCR showing that the Y699F mutant did not alter the induction of pSTAT5 target genes by endogenous pSTAT5 following TPO stimulation (100 ng/ml for 30 min).

K   Quantitative RT–PCR showing that the Y699F mutant repressed the induction of uSTAT5 target genes following TPO treatment.

Data information: Bar graphs represent mean ± SD from three independent experiments. Two-tailed Student's *t*-test; *$P < 0.05$; **$P < 0.01$; ***$P < 0.001$.
Source data are available online for this figure.

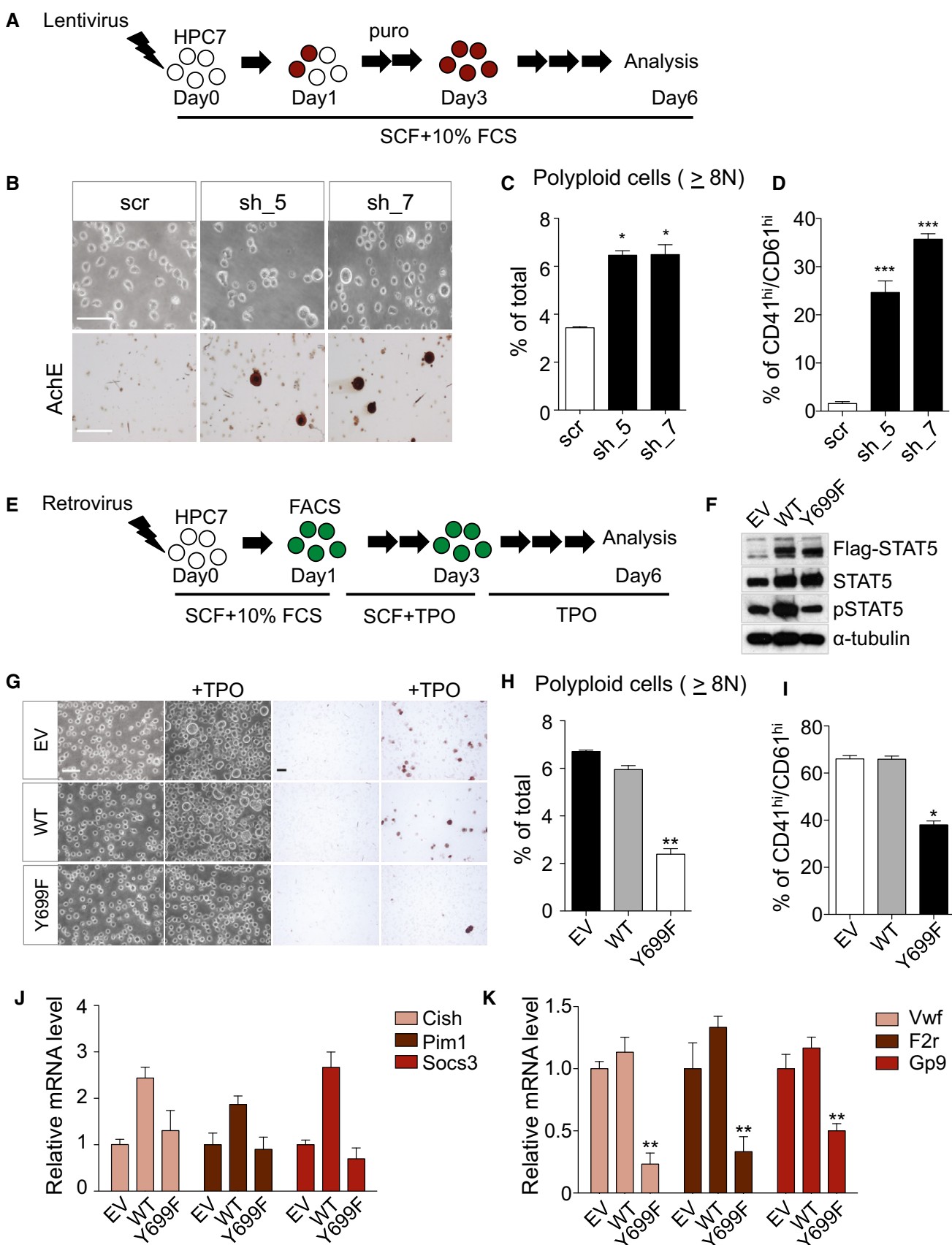

**Figure 4.**

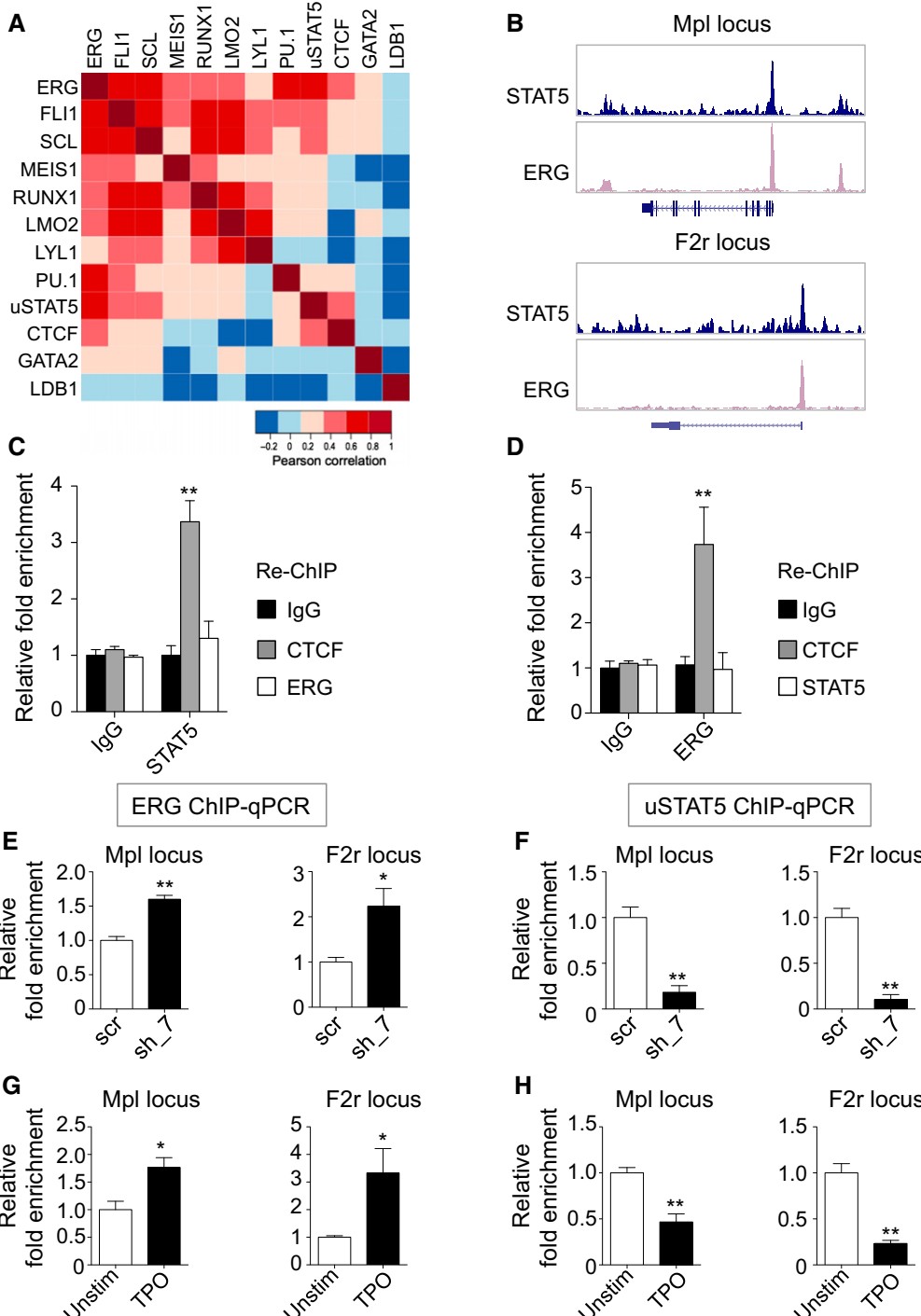

**Figure 5.  uSTAT5 binding highly colocalizes with ERG binding.**

A Heatmap showing hierarchical clustering of pairwise peak overlap of HPC7 ChIP-Seq data, red representing positive Pearson's correlation coefficient values and blue representing negative correlation coefficients.

B Examples of ChIP-Seq genome browser views showing colocalization of uSTAT5 binding with ERG.

C ChIP-reChIP experiment using HPC7 cells grown in SCF-containing media. An antibody to total STAT5 or control IgG was used for the first pull-down and an antibody to CTCF or ERG was used for the second pull-down.

D ChIP-reChIP experiment using HPC7 cells grown in SCF-containing media. An antibody to ERG or control IgG was used for the first pull-down and an antibody to CTCF or STAT5 was used for the second pull-down.

E–H ChIP-qPCR showing an increased enrichment of ERG binding sites at Mpl or F2r loci following STAT5 knockdown (E) or stimulation with TPO (G). ChIP-qPCR showing decreased uSTAT5 binding at Mpl and F2r loci after STAT5 knockdown (F) or following TPO treatment (H).

Data information: All bar graphs represent mean ± SD; $n = 3$; two-tailed Student's $t$-test; *$P < 0.05$; **$P < 0.01$.

 

existence of an uSTAT5 program which represses megakaryocyte-affiliated genes and which is alleviated by TPO-stimulated phosphorylation of STAT5.

Two aspects of our results are of particular significance. Firstly, they reveal a novel mechanism whereby a lineage-specific cytokine modulates JAK/STAT signaling and thus triggers a lineage-affiliated transcriptional program. uSTAT5 colocalizes with three megakaryocytic transcription factors (ERG, FLI1, and SCL) and, at megakaryocytic loci, TPO triggers a loss of uSTAT5 accompanied by increased ERG recruitment. ERG and FLI1 have overlapping essential functions in the formation of megakaryocytes and platelets (Hart *et al*, 2000; Loughran *et al*, 2008; Kruse *et al*, 2009) and play a central role in the transcriptional network that regulates megakaryopoiesis. Coordinate binding of a heptad of transcription factors (including ERG and FLI1) has been demonstrated in HPC7 cells (Wilson *et al*, 2010), and regions bound by the heptad, which contained GATA and ETS sites, were associated with primed or low-level expression of megakaryocytic genes in murine hematopoietic stem/progenitor cells, with GATA2 replaced by GATA1 at a subset of loci during megakaryocytic differentiation (Pimkin *et al*, 2014). Interestingly, we did not identify GATA-binding sites in the vicinity of uSTAT5 peaks, suggesting that the regulatory elements bound by uSTAT5 are distinct from those previously shown to bind GATA1 and GATA2.

Secondly, we demonstrate that uSTAT5 unexpectedly colocalizes with CTCF, thus revealing a potential link between JAK/STAT signaling and nuclear topology. CTCF is an architectural protein that mediates interactions between distant genomic sites, frequently in collaboration with cohesin. As a consequence, it influences transcription by modulating promoter–enhancer interactions (Ong & Corces, 2014), transcriptional pausing (Stadhouders *et al*, 2012; Paredes *et al*, 2013), and alternative splicing (Shukla *et al*, 2011). Moreover, CTCF and cohesin play a key role in compartmentalizing the genomes of higher eukaryotes into functional domains (Nora *et al*, 2012; Seitan *et al*, 2013; Sofueva *et al*, 2013; Zuin *et al*, 2014). It is not yet clear how uSTAT5 is recruited to a specific subset of CTCF-binding sites. Co-immunoprecipitation studies using endogenous or exogenous proteins did not identify an interaction between uSTAT5 and CTCF (data not shown), and it therefore seems likely that uSTAT5 is recruited by one of the multiple proteins known to interact with CTCF (Ong & Corces, 2014).

Previously published STAT5 ChIP-Seq studies did not provide deep sequencing data for the unstimulated condition. However Hirahara *et al* (2015) recently reported binding profiles for STAT3 in unstimulated T cells, as well as T cells stimulated with IL-6 and IL-27. Of note, they showed in their Fig 6A that over 900 peaks are present in the unstimulated condition and go away with both IL-6 and IL-27 stimulation. The authors of this paper make no comment about these peaks and do not investigate them further. In the context of our paper, it seemed likely that at least some of these peaks will correspond to binding by tyrosine-unphosphorylated STAT3 and may therefore not show enrichment of the GAS STAT consensus-binding site. To investigate this further, we downloaded and reanalyzed their raw data. As already noted by Hirahara *et al*, approximately 60% of STAT3-occupied regions showed no change following IL-6 or IL-27 stimulation. The remaining peaks showed dynamic changes in STAT3 occupancy (Appendix Fig S2). Importantly, 902 peaks were specific to unstimulated cells

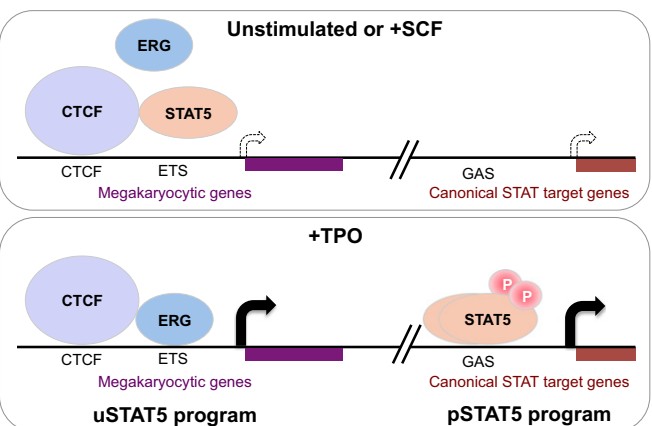

**Figure 6.** Tyrosine-unphosphorylated STAT5 represses a thrombopoietin-inducible megakaryocytic transcriptional program.
Model illustrating the dual STAT5-mediated transcriptional consequences of TPO stimulation: the loss of repression by uSTAT5 of a megakaryocytic program with the concomitant activation of canonical pSTAT target genes.

(Appendix Fig S2, cluster 4). Moreover, peaks specific to unstimulated cells show much lower prevalence of the GAS motif than those peaks appeared after cytokine stimulation. This analysis therefore is consistent with uSTAT3 binding in primary hematopoietic cells and suggests that our model applies to other Stat proteins in a different cell lineage.

Our results also have broad implications for both normal hematopoiesis and the pathogenesis of leukemia. Hematopoietic stem/progenitor cells are characterized by promiscuous expression of genes affiliated to multiple lineages, a phenomenon termed lineage priming (Cross & Enver, 1997; Enver & Greaves, 1998), and it has recently been reported that a subset of hematopoietic stem cells is primed to undergo megakaryocytic differentiation (Sanjuan-Pla *et al*, 2013; Yamamoto *et al*, 2013). We demonstrate that uSTAT5 functions as a transcriptional switch, which mediates TPO-induced upregulation of a poised megakaryocytic program. Multiple hematological malignancies harbor somatic mutations, which increase JAK/STAT signaling (Chen *et al*, 2012; Vainchenker & Constantinescu, 2013); our data indicate that expression of uSTAT5 target genes will be modulated by such mutations and are likely to contribute to the neoplastic phenotype. Increased expression of ERG causes myeloid leukemias in mice (Goldberg *et al*, 2013) and is associated with poor outcome in human AML (Tursky *et al*, 2015); genome-wide loss of uSTAT5 binding may therefore contribute to the leukemogenic consequences of JAK/STAT activation by increasing access of ETS factors to their target genes. Lastly, the discovery of a uSTAT5 transcriptional program has major implications for the development of therapeutic STAT5 inhibitors, most of which currently target only pSTAT5 (Furqan *et al*, 2013).

## Materials and Methods

### Isolation of LSK cells

Bone marrow cells were collected from 6- to 12-week-old C57BL/6 mice. Lineage-positive cells were depleted by using EasySep™

Mouse Hematopoietic Progenitor Cell Enrichment Kit (STEMCELL Technologies). The remaining cells were stained with antibodies against PE-conjugated Sca-1 and APC-cy7-conjugated c-Kit (eBioscience) followed by fluorescence-activated cell sorter (FACS) purification.

## HPC7 cell culture

HPC7 cells were grown in IMDM (Gibco-Life Technologies) supplemented with 10% fetal calf serum (FBS), 10% SCF-conditioned media, 1% L-glutamine, 1% penicillin/streptomycin, and 74.8 μM MTG (Sigma) (Pinto do *et al*, 1998). For serum starvation and TPO stimulation, cells were counted, spun down at 1,000 rpm for 5 min, washed twice in PBS, resuspended at $1 \times 10^6$ cells/ml in StemSpan SFEM (STEMCELL Technologies), and incubated for 4 h. To activate STAT5, we used recombinant murine thrombopoietin (TPO, Peprotech) at 100 ng/ml for 30 min unless indicated otherwise.

## Megakaryocytic differentiation

HPC7 cells were resuspended at $5 \times 10^5$ cells/ml in CellGro media (CellGenix) containing SCF (20 ng/ml) and TPO (100 ng/ml) for 2 days, and then SCF was removed, while TPO remained for an additional 5 days (Dumon *et al*, 2012). For AchE staining, cells were adhered to glass slides by cytospin and stained using AchE staining kit (MBL International) according to the manufacturer's instruction. For ploidy analysis, cells were harvested and fixed with 70% ethanol. The cells were washed twice with PBS and incubated in PI/RNase staining buffer (BD Bioscience) containing 0.1% Triton X-100 for 45 min. The stained samples were analyzed immediately on Cyan flow cytometer. CD41- and CD61-positive cells were determined by flow cytometry performed on Cyan flow cytometer and analyzed using FlowJo software (Tree Star). Antibodies used were FITC- or PB-conjugated CD41 and PE- or APC-conjugated CD61 (eBioscience).

FACS-sorted mouse bone marrow LSK cells ($5 \times 10^3$ cells/ml) were cultured in StemSpan SFEM (STEMCELL Technologies) supplemented with SCF (100 ng/ml) and TPO (100 ng/ml) for 2 days and then in StemSpan SFEM supplemented with SCF (20 ng/ml) and TPO (100 ng/ml) for additional 7 days.

## Chromatin immunoprecipitation sequencing (ChIP-Seq)

Chromatin immunoprecipitation (ChIP) assays were performed as previously described (Forsberg *et al*, 2000; Wilson *et al*, 2010) using antibodies against total STAT5 (Santa Cruz, sc835), ERG (Santa Cruz, sc354x), CTCF (Millipore, 07-592), and control rabbit IgG (Invitrogen, 9172). ChIP DNA samples were amplified for sequencing with the Illumina Chip-Seq Sample Prep Kit (Illumina) and sequenced with the Illumina GA2 Genome Analyser platform. The ChIP-Seq data from this study have been submitted to the NCBI Gene Expression Omnibus (GEO; http://www.ncbi.nlm.nih.gov/geo/) under accession number GSE70697.

## Immunofluorescence assay

One-third of FACS-sorted LSK cells were directly spun onto the glass slides (freshly isolated). Remaining cells were attached to glass slides by cytospin following incubation in StemSpan SFEM (STEMCELL Technologies) for 4 h (Starved) and subsequent TPO (100 ng/ml) stimulation for 30 min (TPO treated). Staining for total STAT5 (Santa Cruz, sc835) and phospho-STAT5 (Cell Signaling, 9314) was carried out after fixation and permeabilization. The cell nuclei were counterstained with 4′,6-diamidino-2-phenylindole (DAPI). To quantify fluorescence, at least 10 fields were randomly selected and quantified on image analyzer (Image J, NIH). Briefly, each nucleus was outlined from thresholded images of the DAPI channel and this was then superimposed onto the corresponding fluorescent image of the same nucleus. In each nucleus, the average pixel intensity (API) per unit area was calculated. The same outline of each nucleus cone was then moved onto an adjacent empty region, and the API of this background was calculated. The API of each background was subtracted from the API of nucleus to calculate the corrected nuclear API.

## Western blotting

Whole-cell lysates were prepared by resuspending cells in modified RIPA buffer and 2× Laemmli sample buffer (Bio-Rad). Protein samples were separated by 8% SDS–PAGE and transferred to nitrocellulose membranes (Millipore). The membranes were blocked in TBST containing 3% non-fat milk prior to incubation with primary antibodies. Detection was performed with horseradish peroxidase (HRP)-conjugated secondary antibody (BioRad) and enhanced by chemiluminescence (ECL) reagent (Pierce). Image J software was used to analyze the densitometry value of Western blots bands.

## Cell fractionations

Nuclear and cytoplasmic fractions were prepared as previously described (Osborn *et al*, 1989). Briefly, cells were washed twice with PBS and once in Buffer A (10 mM HEPES pH 7.9, 1.5 mM $MgCl_2$, 10 mM KCl, 0.5 mM DTT and protease inhibitor cocktail). Cells were then pelleted and resuspended in Buffer A with 0.1% NP-40 and incubated on ice for 10 min. The supernatant containing the cytoplasmic fraction was collected by centrifugation, and the pellet was resuspended in an equal volume (relative to the cytoplasmic extract) of Buffer B (20 mM HEPES pH 7.9, 1.5 mM $MgCl_2$, 300 mM NaCl, 0.5 mM DTT, 25% glycerol, 0.2 mM EDTA, and protease inhibitor cocktail). To extract chromatin-bound proteins, Pierce Universal Nuclease (Thermo Scientific) was added to the nuclear lysate and incubated for 90 min on ice. Following centrifugation, equal volumes of nuclear and cytoplasmic fractions were analyzed by immunoblotting.

## RNA-Seq and quantitative RT–PCR

HPC7 cells were infected with lentivirus containing shRNA specific for STAT5. Scrambled shRNA was used as mock controls. Infected cells were selected using puromycin (1 μg/ml) for 2 days and recovered in IMDM supplemented with 10% FBS and 10% SCF-conditioned media for 24 h. Total RNA was isolated using RNeasy kit (Qiagen) and subjected to RNA sequencing (BGI). The gene expression level was calculated by using RPKM method (Reads Per kb per Million reads). DEseq packages (Wang *et al*, 2010) were used for standardization and identification of differentially expressed

genes (Anders & Huber, 2010). We defined the differentially expressed genes that satisfy FDR <0.001 and fold change >1.5. Lineage signatures were analyzed by GSEA (Subramanian *et al*, 2005) using published microarray data of different hematopoietic progenitor cells (Pronk *et al*, 2007). For RT–PCR, total RNA was reverse-transcribed into first-strand cDNA using SuperScriptIII First Strand Synthesis Super Mix (Invitrogen) following the manufacturer's instructions. Quantitative RT–PCR was carried out using SYBR Green Brilliant II low Rox on Stratagene Mx3000P (Agilent Technologies).

### STAT5 retrovirus expression

Retrovirus was produced using the pCL-Eco Retrovirus Packaging Vector (Imgenex). HPC7 cells were infected with retrovirus in the presence of 8 µg/ml polybrene (Millipore). After 24 h, virus-infected GFP-positive cells were FACS-sorted and cultured for another 48 h in IMDM supplemented with 10% FBS and 10% SCF-conditioned media before megakaryocyte differentiation.

### STAT5 knockdown

Two independent lentiviral shRNAs (TRCN0000012553 and TRCN0000012557) were used to knockdown STAT5B protein. Lentivirus was produced by transfecting 293T cells with 7 µg/ml STAT5 shRNA plasmids, 5 µg/ml psPAX2, and 2 µg/ml pMD2.G. Forty-eight hours after 293T cell transfection, supernatant containing lentivirus was collected and concentrated using Lenti-X concentrator (Clontech). For infection, 100 µl of concentrated lentivirus was added to each well of a 6-well plate containing $1 \times 10^5$ cells. Cells were incubated with lentivirus for 24 h and subjected to puromycin selection (1–2 µg/ml) for 2 days. Control cells were infected with lentivirus containing scrambled shRNA. Quantification of the Western blot images with Image J showed the STAT5-knockdown efficiency in HPC7 cells is higher than 95% at protein level.

### Statistical analysis

The results are presented as means ± SD. Experimental data were analyzed using two-tailed Student's *t*-test. *P*-values less than 0.05 were considered statistically significant.

### ChIP-Seq analysis

#### Peak calling and heatmap
Significant binding events in comparison with a control IgG sample were found using MACS (Zhang *et al*, 2008). Peak coordinates in regions with high repeat content were subtracted from the list of high confidence peak coordinates. All peaks were standardized to 400 bp based on MACS summits. The heatmap was generated in R using the heatmap.2 function of the gplots package, with hierarchical clustering using the "ward" method. Colors were generated using the package RColorBrewer.

#### Motif analysis
*De novo* motif finding analysis was conducted with MEME (Bailey *et al*, 2009) using standard settings. We used the sequences of 100 bp around the peak summit as input for the motif discovery.

Peaks containing more than 40% of repetitive sequences were excluded from the analysis. Matches to consensus sequences were determined using TOMTOM (Gupta *et al*, 2007).

Central motif enrichment analysis (CMEA) was performed using CentriMo (Bailey & Machanick, 2012). This algorithm has been developed based on the observation that the direct TF (transcription factor) binding sites tend to cluster near the center of any ChIP-ed regions.

#### Gene mapping
Peaks were mapped to the nearest gene first considering those whose promoter overlapped the peak, then those genes directly overlapping peaks in other regions, and then any gene within 50 kb on both directions of the peak coordinates (simultaneous presence of genes on both the 5′ and 3′ flanking window was resolved considering both genes). We used gene coordinates as defined by UCSC and promoter as defined on the mammalian promoter database (MPromDB).

**Expanded View** for this article is available online.

### Acknowledgements
We thank Charlie Massie, Felicia Ng, Stephen Loughran, and Winnie Lau for discussions and Federico Comoglio for comments on this manuscript. Work in the Green lab is supported by Bloodwise (grant ref. 13003), the Wellcome Trust (grant ref. 104710/Z/14/Z), the Medical Research Council, the Kay Kendall Leukaemia Fund, the Cambridge NIHR Biomedical Research Center, the Cambridge Experimental Cancer Medicine Centre, the Leukemia and Lymphoma Society of America (grant ref. 07037), and a core support grant from the Wellcome Trust and MRC to the Wellcome Trust-Medical Research Council Cambridge Stem Cell Institute. Hyun Jung Park was supported by postdoctoral fellowships from the EMBO and the Human Frontier Science Program.

### Author contributions
ARG and BG directed the work. HJP, with the help of ARG, BG, and VS, wrote the manuscript. HJP and JL performed experiments and analysis. RH, SB, and DFSC analyzed the ChIP-Seq data. AIL-C and KK helped with experiments.

### Conflict of interest
The authors declare that they have no conflict of interest.

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
