## [Review Process File · The EMBO Journal]

Manuscript EMBO-2015-92383

Cytokine-induced megakaryocytic differentiation is regulated by genome-wide loss of a uSTAT transcriptional program

Hyun Jung Park , Juan Li , Rebecca Hannah , Simon Biddie , Ana Leal-Cervantes , Dr. Kristina Kirschner , David Flores Santa Cruz , Prof. Veronika Sexl , Prof. Berthold Göttgens and Anthony R. Green

Corresponding author: Anthony R. Green, University of Cambridge

Review timeline:

Submission date:	29 June 2015
Editorial Decision:	13 August 2015
Editorial Decision:	02 September 2015
Revision received:	24 October 2015
Editorial Decision:	19 November 2015
Revision received:	27 November 2015
Accepted:	01 December 2015

Editor: Bernd Pulverer

Transaction Report:

Preliminary Editorial Decision

13 August 2015

Thank you for submitting your manuscript for consideration by the EMBO Journal. Please find enclosed the comments of two of the three reviewers whom we had asked to evaluate the manuscript. We are still waiting for a third report but given the well aligned recommendation of the two reports in hand, despite the lack of detail provided in the second report, I am sending a preliminary decision now to avoid further delays. This decision is still subject to change should the third referee offer strong and compelling reasons for doing so (and assuming the report arrives before a resubmission in sent). As you will see, the two present respondents find the topic of your manuscript interesting in principle.

As we discussed, we share their interest in principle and would like to encourage a revision based on the important issues raised by referee 1. This includes a detailed characterization of the Stat5 AB, ChIP-Seq in Stat5 null cells and unstimulated primary hematopoietic cells and investigation of Socs2 and Cish. Please also address point 11.

We will forward the comments of the third referee to you as soon as we receive them, together with our final editorial decision. I am sorry for these delays, unfortunately due to travel schedules by the referees and editors.

REFeree REPORTS

Referee #1:

The manuscript by Park and colleagues describes novel findings that are of great relevance to the understanding of STAT transcription factors in the establishment of specific cell lineages. While there has been great emphasis on cytokine-induced activation of STAT5 (pSTAT5) and its role in executing cell-specific genetic programs, potential roles of unstimulated STAT (uSTAT5) had not been studied. The authors made the unexpected observation that uSTAT5 binds to sites also occupied by CTCF, a protein suspected to control chromatin looping and by that token also cell-specific gene control. With the stimulation of cells by TPO there is a major rearrangement of STAT5 binding to sites containing GAS motifs, which is accompanied by the expected activation of TPO-inducible genes.

This work has the potential of becoming very influential in our understanding of how cells progress through lineage-specific differentiation programs.

Critique

1. What is the evidence that the AB specifically recognizes only STAT5? Have the authors tested the AB in bona fide Stat5-null cells, both for IF and ChIP-seq? ShRNA-based knock-down might not be sufficient because as little as 10% STAT5 is sufficient for cells to respond to cytokines and establish normal STAT5 occupancy as measured by ChIP-seq.
2. The specificity of total STAT5 antibody and the reliability of uSTAT5 ChIP-Seq are of concern to this reviewer. ChIP-Seq using total STAT5 antibody should be also performed in bona fide Stat5-null cells (or >95% knock-down) to identify the "real" binding sites of uSTAT5.
3. The so-called "uSTAT5 target genes" and "pSTAT5 target genes" are misleading. In Fig. 2 and S2, the authors compared the STAT5 ChIP-Seq data in untreated cells and 30min TPO-stimulated cells, presenting uSTAT5 and pSTAT5 binding profiles respectively. However, it seems that pSTAT5 only binds to general target genes, Cish, Socs2, BCL6, etc., at this early point 30 min. It also resulted in very limited number of detectable pSTAT5 binding peaks (only 994 peaks in Fig. 2F). Given to the proposed model, pSTAT5 binding to lineage-specific genes is expected. Does pSTAT5 bind and activate Mpl and F2r in differentiated cells? Instead of 30min, the authors should perform pSTAT5 ChIP-Seq at day 7 and re-analyze the data in Fig. 2A and F.
4. What is the percentage of uSTAT5 in TPO-stimulated HPC7 cells?
5. Have the authors tried to conduct ChIP-seq experiments in unstimulated primary hematopoietic cells?
6. Have the authors analyzed data sets from unstimulated T cells (available on GEO)?
7. It would be helpful if the authors could provide some close-up screen shots of areas binding CTCF and uSTAT5.
8. In Figure 3, the authors conduct STAT5 knock-down experiments. What is the percent reduction of STAT5? See my point #1
9. How do bona fide STAT5 targets, such as Socs2 and Cish, respond to the knock-down?
10. What happens to "pSTAT5 target genes" upon STAT5 knockdown in unstimulated cells? The data could be provided along with Fig. 3D as control.
11. In Fig. 4, the mechanism of defective differentiation in Y669F mutant over-expressed cells is not clear yet. As pointed out earlier, "uSTAT5 target genes" may also be "pSTAT5 target genes" if late stage ChIP-Seq is performed. Although protein levels of pSTAT5 seems unaltered (Fig. 4F), the mutant could affect the dimerization of pSTAT5 and/or pSTAT5 binding to target genes, perhaps including F2r, Gp9 etc.
12. The GEO/GSE numbers of the ChIP-seq experiments need to be provided so reviewers and readers have the opportunity to evaluate the primary data, which is essential for studies that provide new, and ground-breaking, concepts.

Referee #2:

I have no critical comment to offer. The experiments have been well designed and their interpretation is appropriate. The writing is excellent. There are of course questions about how uSTAT5 binds to and dissociates from DNA, but the authors describe well their future plans to address these issues in the discussion.

1st Editorial Decision

02 September 2015

We have now obtained the third referee report, which affirms the positive views of the other two referees.

The referee suggests a couple of textural issues which should be addressed. If you have available data of Y699F in a Stat5b null background, we agree with the referee that this should be added.

Given the three referees' positive recommendations, I would like to reiterate that we invite you to submit a revised version of the manuscript, addressing the comments of all three reviewers. Please refer back to the substantive experimental issues discussed previously as a precondition for a successful revision. I also reiterate that it is EMBO Journal policy to allow only a single round of revision, and acceptance of your manuscript will therefore depend on the completeness of your responses in this revised version

Thank you for the opportunity to consider your work for publication. I look forward to your revision.

REFEREE REPORT

Referee #3:

The report by Park and colleagues shows a comprehensive analysis how Stat5 impacts on TPO-dependent megakaryocyte differentiation. Strikingly, the authors demonstrate an important role for nuclear unphosphorylated Stat5 (uStat5) in antagonizing the Ets family transcription factor ERG, thus inhibiting a transcriptional program favouring megakaryocyte fate. Conversely, tyrosine-phosphorylated Stat5 (pStat5) redistributes to different genomic loci following treatment with TPO. This redistribution is correlated with a GAS element-driven transcriptional program involving typical, not lineage-restricted Stat5 target genes.

The paper provides compelling evidence for the model in fig. 6 which depicts the major conclusions. A minor weakness is the fact that the lentiviral introduction of the Y699F mutant of Stat5b is performed in a wt background. That said the wealth of evidence for the uStat5 impact on transcription appears sufficient.

Specific comments:

1. figure 1E: why is the pStat5 signal in the input sample of TPO-treated cells so low?
2. The CentriMo analysis is poorly explained (I haven't seen any description in the methods section) and confusing: the statement in the legend to S2D that Stat5 peaks are centrally located within pStat5 peaks appears at odds with the redistribution analysis shown in fig. 2A. According to 2G CTCF peaks overlap with or flank uStat5 peaks. The relationship to data in S2D should be clarified.
3. Is there any evidence from published Stat5 ChIP-Seq data in other cells that a similar redistribution occurs upon phosphorylation (discussion point)?

Response to Reviewers

We were delighted to read the unanimously positive comments by all three reviewers. Reviewer #2 was happy with our original submission. Reviewers #1 and #3 made a number of constructive comments, all of which we have addressed as outlined below (reviewers comments in italics).

Referee #1

(Report for Author)

The manuscript by Park and colleagues describes novel findings that are of great relevance to the understanding of STAT transcription factors in the establishment of specific cell lineages. While there has been great emphasis on cytokine-induced activation of STAT5 (pSTAT5) and its role in executing cell-specific genetic programs, potential roles of unstimulated STAT (uSTAT5) had not been studied. The authors made the unexpected observation that uSTAT5 binds to sites also occupied by CTCF, a protein suspected to control chromatin looping and by that token also cell-specific gene control. With the stimulation of cells by TPO there is a major rearrangement of STAT5 binding to sites containing GAS motifs, which is accompanied by the expected activation of TPO-inducible genes.

This work has the potential of becoming very influential in our understanding of how cells progress through lineage-specific differentiation programs.

We thank the reviewer for acknowledging that our paper has the potential to be very influential. The reviewer also made a number of constructive comments, all of which we have addressed by performing a number of additional experiments and revising the paper as outlined below.

Critique

1. What is the evidence that the AB specifically recognizes only STAT5? Have the authors tested the AB in bona fide Stat5-null cells, both for IF and ChIP-seq? ShRNA-based knock-down might not be sufficient because as little as 10% STAT5 is sufficient for cells to respond to cytokines and establish normal STAT5 occupancy as measured by ChIP-seq.

And 2. The specificity of total STAT5 antibody and the reliability of uSTAT5 ChIP-Seq are of concern to this reviewer. ChIP-Seq using total STAT5 antibody should be also performed in bona fide Stat5-null cells (or >95% knock-down) to identify the "real" binding sites of uSTAT5.

We appreciate the reviewer's concern on this point. We have now performed IF and WB using STAT5 null MEFs. These data clearly show that the STAT5 antibody used for this study (sc-835, Santa Cruz Biotechnology) is highly specific to STAT5 proteins. We also performed ChIP-Seq using total STAT5 antibody in STAT5 knock-down HPC7 cells (>95% knock-down). The numbers of raw and uniquely mapped reads are summarized below.

Sample	Reads	Mapped	%
IgG	24,944,125	5,986,179	24.00
STAT5	35,778,166	7,592,581	21.22

We further analyzed the ChIP-Seq data and found very few peaks regions, indicating that uSTAT5 peaks we observed in this study are specific to STAT5 protein. We have added these results to the manuscript and present them in Supplementary Figure S3.

3. The so-called "uSTAT5 target genes" and "pSTAT5 target genes" are misleading. In Fig. 2 and S2, the authors compared the STAT5 ChIP-Seq data in untreated cells and 30min TPO-stimulated cells, presenting uSTAT5 and pSTAT5 binding profiles respectively. However, it seems that pSTAT5 only binds to general target genes, Cish, Socs2, BCL6, etc., at this early point 30 min. It also resulted in very limited number of detectable pSTAT5 binding peaks (only 994 peaks in Fig. 2F).

The reviewer pointed out that the number of pSTAT5 peaks we obtained in this study is small compared to other STATs in other systems. To address this point, we analyzed pSTAT5 ChIP-Seq data sets reported by other groups and found that the number of peaks varied greatly between cell types and treatment conditions from 215 (GSM1234366) to 6624 (GSM904760). We hope that the reviewer agrees that our number of 994 peaks falls within this range.

Given to the proposed model, pSTAT5 binding to lineage-specific genes is expected. Does pSTAT5 bind and activate Mpl and F2r in differentiated cells? Instead of 30min, the authors should perform pSTAT5 ChIP-Seq at day 7 and re-analyze the data in Fig. 2A and F.

The reviewer raised the question of whether pSTAT5 might be present at late time points and be able to activate megakaryocytic genes. To address this question, we have now examined pSTAT5 levels at various time points up to 7 days and found that there is no detectable pSTAT5 by 24 hours (fig. S1E). In addition, we performed the pSTAT5 ChIP-Seq at day 7 as suggested by the reviewer. However, the pSTAT5 antibody had a very low efficiency of immunoprecipitating mappable DNA under these conditions; in fact its efficiency was considerably lower than the non-specific IgG as outlined by the read numbers given below.

Sample	Reads	Mapped	%
IgG	25,680,447	7,763,465	30.23
pSTAT5	50,226,071	1,527,804	3.04

This low efficiency is completely consistent with our Western blot analysis, which showed that there simply isn't any pStat5 around 7 days after Tpo stimulation. The new western blot data have been included (fig. S1E) and the text amended to clarify this point on page 4, line 23.

4. What is the percentage of uSTAT5 in TPO-stimulated HPC7 cells?

To directly address this question, we would need to perform quantitative mass spectrometry. While we agree that it would be interesting to examine the percentage of uSTAT5 in TPO-stimulated cells, we believe that the generation of such data would be beyond of the scope of the current paper, which is already very data-rich. Of note, it has been shown that 73% of STAT5B molecules in erythroid progenitor cells become tyrosine phosphorylated within 10 min of EPO stimulation (1). Given the overlap between EPO and TPO signaling pathways, we speculate that TPO stimulation may result in similar levels of STAT5 phosphorylation.

5. Have the authors tried to conduct ChIP-seq experiments in unstimulated primary hematopoietic cells?

We thank the reviewer and agree that this is an important question. Primary hematopoietic cells directly comparable to the HPC7 cells would be highly-purified common myeloid progenitor cells. This is a rare cell population and it is not possible to obtain sufficient cells for transcription factor ChIP-Seq analysis. We therefore searched the literature for studies in other primary haematopoietic cells that might be informative.

We first analyzed STAT5 ChIP-Seq data from unstimulated T cells available on GEO (GSE41317 and GSE36882). Unfortunately, the samples had poor uniquely mappable alignment rates, and the density plots created from the remaining reads (bigWigs) are very noisy with very few clear peak regions.

However, Hirahara et al (2) recently reported binding profiles for STAT3 in unstimulated T-cells, as well as T-cells stimulated with IL6 and IL27. Of note, they showed in their Figure 6A that over 900 peaks are present in the unstimulated condition and go away with both IL6 and IL27 stimulation. The authors of this paper make no comment about these peaks, and do not investigate them further. In the context of our paper, it seemed likely that at least some of these peaks will correspond to binding by tyrosine-unphosphorylated STAT3, and may therefore not show enrichment of the GAS STAT consensus binding site. To investigate this further, we downloaded and reanalysed their raw data. As already noted by Hirahara et al, approximately 60% of STAT3 occupied regions showed no change following IL6 or IL27 stimulation. The remaining peaks showed dynamic changes in STAT3

occupancy (fig. S8). Importantly, 902 peaks were specific to unstimulated cells (fig. S8, cluster 4). Moreover, peaks specific to unstimulated cells show much lower prevalence of the GAS motif than those peaks appeared after cytokine stimulation. This analysis therefore is consistent with uSTAT3 binding in primary haematopoietic cells and suggests that our model applies to other Stat proteins in a different cell lineage. We have incorporated the new analysis into supplementary figure S8 and amended the text on page 13, line 10.

6. *Have the authors analyzed data sets from unstimulated T cells (available on GEO)?*

Yes, we have now done this. Please see response to question 5.

7. *It would be helpful if the authors could provide some close-up screen shots of areas binding CTCF and uSTAT5.*

The close-up screen shots have now been provided and can be seen in Figure S2E.

8. *In Figure 3, the authors conduct STAT5 knock-down experiments. What is the percent reduction of STAT5? See my point #1*

We have quantified STAT5 WB using ImageJ and found that STAT5 knock-down efficiency was 97.5 and 99.8 % (sh_5 and sh_7, respectively). We refer to this in the revised methods section.

9. *How do bona fide STAT5 targets, such as Socs2 and Cish, respond to the knock-down? And 10. What happens to "pSTAT5 target genes" upon STAT5 knockdown in unstimulated cells? The data could be provided along with Fig. 3D as control.*

We performed RNA-Seq experiment using self-renewing HPC7 cells, which do not express detectible levels of pSTAT5. As expected, we observed no significant change in transcript levels of the bona fide pSTAT5 targets, including Socs2 and Cish, following STAT5 knock-down. We now refer to this observation in the revised manuscript on page 7, line 19.

11. *In Fig. 4, the mechanism of defective differentiation in Y669F mutant over-expressed cells is not clear yet. As pointed out earlier, "uSTAT5 target genes" may also be "pSTAT5 target genes" if late stage ChIP-Seq is performed.*

Although protein levels of pSTAT5 seems unaltered (Fig. 4F), the mutant could affect the dimerization of pSTAT5 and/or pSTAT5 binding to target genes, perhaps including F2r, Gp9 etc.

The possibility that those specific uSTAT5 target genes may also become pSTAT5 targets at a later time point has now been addressed by our new data (Please response to point #3).

Our data show that the Y694F mutant reduces expression of megakaryocyte-affiliated uSTAT5 target genes (fig. 4K), but does not reduce expression of canonical pSTAT5 target genes (fig. 4J). The later observation indicates that Y694F does not affect the function of pSTAT5, including dimerization of pSTAT5 and/or pSTAT5 binding to those genes. We have amended the text to clarify this point (page 9, line 18).

12. *The GEO/GSE numbers of the ChIP-seq experiments need to be provided so reviewers and readers have the opportunity to evaluate the primary data, which is essential for studies that provide new, and ground-breaking, concepts.*

We thank the reviewer for pointing it out. We have now provided GSE number (GSE70697) for the ChIP-Seq experiments in the text.

Referee #2

(Report for Author)

I have no critical comment to offer. The experiments have been well designed and their interpretation is appropriate. The writing is excellent. There are of course questions about how

uSTAT5 binds to and dissociates from DNA, but the authors describe well their future plans to address these issues in the discussion.

We were delighted to read this positive feedback. The reviewer did not request any revisions to be carried out.

Referee #3:

The report by Park and colleagues shows a comprehensive analysis how Stat5 impacts on TPO-dependent megakaryocyte differentiation. Strikingly, the authors demonstrate an important role for nuclear unphosphorylated Stat5 (uStat5) in antagonizing the Ets family transcription factor ERG, thus inhibiting a transcriptional program favouring megakaryocyte fate. Conversely, tyrosine-phosphorylated Stat5 (pStat5) redistributes to different genomic loci following treatment with TPO. This redistribution is correlated with a GAS element-driven transcriptional program involving typical, not lineage-restricted Stat5 target genes. The paper provides compelling evidence for the model in fig. 6 which depicts the major conclusions. A minor weakness is the fact that the lentiviral introduction of the Y699F mutant of Stat5b is performed in a wt background. That said the wealth of evidence for the uStat5 impact on transcription appears sufficient.

We thank this reviewer for his/her positive assessment of our paper, and were particularly pleased read the comment that “*the wealth of evidence for the uStat5 impact on transcription appears sufficient*“. The reviewer also made a number of constructive comments, which we have addressed as outlined below.

Specific comments:

1. figure 1E: why is the pStat5 signal in the input sample of TPO-treated cells so low?

The reason is that we loaded only 1% of input (5µg). We have now clarified this in the figure legend.

2. The CentriMo analysis is poorly explained (I haven't seen any description in the methods section) and confusing: the statement in the legend to S2D that Stat5 peaks are centrally located within pStat5 peaks appears at odds with the redistribution analysis shown in fig. 2A. According to 2G CTCF peaks overlap with or flank uStat5 peaks. The relationship to data in S2D should be clarified.

We performed Central Motif Enrichment Analysis (CMEA) using CentriMo (3). This algorithm has been developed based on the observation that the direct TF (transcription factor) binding sites tend to cluster near the center of any ChIP-ed regions.

Figure S2D shows that the “STAT5 consensus binding motif” was found in the centre of pSTAT5 peaks. The *mpl* locus shown in fig. 2G is a genomic region spanning approximately 40 kb (chr4:118,106,637-118,141,172), whereas the CentriMo analysis was done on STAT5 peak regions (400 bp), as it is designed to be used in this way. Figure S2D shows that the CTCF motif was found in the centre of the peak regions.

We apologize if there was a lack of clarity in the original manuscript and have now clarified this point in both the Methods section and Figure Legends.

3. Is there any evidence from published Stat5 ChIP-Seq data in other cells that a similar redistribution occurs upon phosphorylation (discussion point)?

Yes, we now provide such evidence; please see response to reviewer 1's point #6

References

1. J. Bachmann *et al.*, Division of labor by dual feedback regulators controls JAK2/STAT5 signaling over broad ligand range. *Mol Syst Biol* 7, 516 (2011).

2. K. Hirahara *et al.*, Asymmetric Action of STAT Transcription Factors Drives Transcriptional Outputs and Cytokine Specificity. *Immunity* **42**, 877-889 (2015).
3. T. L. Bailey, P. Machanick, Inferring direct DNA binding from ChIP-seq. *Nucleic Acids Res* **40**, e128 (2012).

2nd Editorial Decision

19 November 2015

Thank you for re-submitting your revised manuscript. It has now been seen by referee1, who is in favour of publication. We will be happy to pursue rapid publication pending satisfactory minor revision as outlined below:

- Please note the in fig 1c, the pSTAT5 (Y694/699) panel contains an unexplained shadow - this may be a photoshop artefact. Also please note that the control band from K562 cells is fairly similar to fig S3A, w.t. - both contain a strange dot on the top right and the morphology of the main band is somewhat similar. Please ensure there is no duplication. Please provide source data where possible (see below).
- we require individual files for each figure.
- we require scale bars for fig 4B, 4G, S3B.
- please define the stats in Fig 3E, 3F, S4C, S5C, S6F
- the materials and methods section has to be integrated into the main manuscript
- we only allow supplementary figures where absolutely necessary and instead integrate figures in a collapsible manner into the main manuscript as fully copyedited, integral figures called 'Expanded View (EV)'. We recommend that current S1, S2, S4, S5, S6 are elevated to 'Expanded View' figures in line in the main manuscript. Current Table S1, and S7 can be retained as supplementary figures. Current S3 should in our view be integrated into the existing figures.
- we require author conflict of interest statements (which will be published)
- we require an author contribution statement
- we can only formally accept a manuscript with a satisfactory author checklist that meets EMBO Journal publication guidelines.
- we acknowledge that a synopsis blurb is included (in the text after abstract). Please add 3-5 bullet points as per house style for synopses.
- Please add source data to figures where possible for Western blot IPs, graphs and micrographs. Replicates can by all means be included.
- Please review the title and abstract carefully to ensure that all keywords are included to optimize discoverability. We suggest to include 'megakaryocyte' in the title. Maybe use a more precise verb than 'reflects' in the title. Please define TPO at first use (abstract). CTCF could be labelled as a TF for the general reader.

Thank you for the opportunity to publish your work. I look forward to your revision.

REFEREE REPORT

Referee #1:

I looked over the manuscript and the response and the authors have addressed my concerns.

Response to Editor

1. Please note the in fig 1c, the pSTAT5 (Y694/699) panel contains an unexplained shadow -this may be a photoshop artefact. Also please note that the control band from K562 cells is fairly similar to fig S3A, w.t. -both contain a strange dot on the top right and the morphology of the main band is somewhat similar. Please ensure there is no duplication. Please provide source data where possible (see below).

We had a close look at the original image of fig 1c. We agree that the shadow might be a photoshop artefact and it seems that the original image was not very good quality. We therefore have repeated the western blotting and included in the figure 1 and the source data for figure 1.

Also, we have compared the two blots (fig1c and figS3A) and confirmed that they are not the same. Please see the attached images. To avoid further confusion, we have changed them with the replicate experiments.

2. we require individual files for each figure.

We will upload individual files for each figure.

3. we require scale bars for fig 4B, 4G, S3B.

We have included scale bars for the figures.

4. please define the stats in Fig 3E, 3F, S4C, S5C, S6F

We have defined the stats as requested.

5. the materials and methods section has to be integrated into the main manuscript

We have integrated the materials and methods into the main manuscript.

6. we only allow supplementary figures where absolutely necessary and instead integrate figures in a collapsible manner into the main manuscript as fully copyedited, integral figures called 'Expanded View (EV)'. We recommend that current S1, S2, S4, S5, S6 are elevated to 'Expanded View' figures in line in the main manuscript. Current Table S1, and S7 can be retained as supplementary figures. Current S3 should in our view be integrated into the existing figures.

We have made the changes as suggested.

7. we require author conflict of interest statements (which will be published)

We have included the author conflict of interest statements in the manuscript. Please see page 21.

8. we require an author contribution statement

We have included the author contribution statement in the manuscript. Please see page 21.

9. we can only formally accept a manuscript with a satisfactory author checklist that meets EMBO Journal publication guidelines.

We have completed the author checklist.

10. we acknowledge that a synopsis blurb is included (in the text after abstract). Please add 3-5 bullet points as per house style for synopses.

We have included the synopsis in the manuscript.

11. Please add source data to figures where possible for Western blot IPs, graphs and micrographs. Replicates can by all means be included.

We are happy to provide the source data.

12. Please review the title and abstract carefully to ensure that all keywords are included to optimize discoverability. We suggest to include 'megakaryocyte' in the title. Maybe use a more precise verb than 'reflects' in the title. Please define TPO at first use (abstract). CTCF could be labelled as a TF for the general reader.

We have amended the title and abstract as requested.

3rd Editorial Decision

1 December 2015

I am pleased to inform you that your manuscript has been accepted for publication in the EMBO Journal. We will proceed with rapid publication and my colleagues will request any additional information separately.

Corresponding Author Name: Anthony R Green

Manuscript Number: EMBOJ-2015-92383R